# Discoveries for Long Non-Coding RNA Dynamics in Traumatic Brain Injury

**DOI:** 10.3390/biology9120458

**Published:** 2020-12-10

**Authors:** Key-Hwan Lim, Sumin Yang, Sung-Hyun Kim, Sungkun Chun, Jae-Yeol Joo

**Affiliations:** 1Neurodegenerative Disease Research Group, Korea Brain Research Institute, Daegu 41062, Korea; khlim@kbri.re.kr (K.-H.L.); suminyang.v@kbri.re.kr (S.Y.); butterfly224@kbri.re.kr (S.-H.K.); 2Department of Physiology, Jeonbuk National University Medicine School, Jeonju 54907, Korea; sungkun.chun@jbnu.ac.kr

**Keywords:** biomarker, long non-coding RNA, non-degenerative disease, traumatic brain injury

## Abstract

**Simple Summary:**

The biomedical studies of traumatic brain injury (TBI) can lead to insight for treatment clinically. However, TBIs are occurred by various risk factors and showing heterogeneity that make difficult to accurate diagnosis for initiation treatment of patients. Therefore, identification of biomarkers requires to prediction and therapeutics for TBI treatment. The canonical function of the long non-coding RNAs (lncRNAs) have been recently shown to promote transcription, post-transcription, and protein activity in many different conditions. Therefore, understanding the molecular mechanisms that are altered by the expression of lncRNAs will allow the design of novel therapeutic strategies. Here, we review the molecular process of lncRNA as new targets and approaches in TBIs treatment.

**Abstract:**

In recent years, our understanding of long non-coding RNAs (lncRNAs) has been challenged with advances in genome sequencing and the widespread use of high-throughput analysis for identifying novel lncRNAs. Since then, the characterization of lncRNAs has contributed to the establishment of their molecular roles and functions in transcriptional regulation. Although genetic studies have so far explored the sequence-based primary function of lncRNAs that guides the expression of target genes, recent insights have shed light on the potential of lncRNAs for widening the identification of biomarkers from non-degenerative to neurodegenerative diseases. Therefore, further advances in the genetic characteristics of lncRNAs are expected to lead to diagnostic accuracy during disease progression. In this review, we summarized the latest studies of lncRNAs in TBI as a non-degenerative disease and discussed their potential limitations for clinical treatment.

## 1. Introduction

Transcription is a fundamental biological process, and the genome is transcribed into two distinct types of RNAs, namely, protein-coding RNAs and non-coding RNAs (ncRNAs) [1]. While mRNAs become templates to translate the genetic information, most ncRNAs, such as microRNAs (miRNAs), small nucleolar RNAs (snoRNAs), and enhancer RNAs (eRNA), and long non-coding RNAs (lncRNAs) do not have the potential to encode functional proteins [2,3,4]. Long non-coding RNAs (lncRNAs) are transcribed by RNA polymerase II (Pol II) and are longer than 200 nucleotides (nt) (Figure 1). A recent study using total RNA sequencing (RNA–seq) revealed that over 58,000 lncRNAs are encoded in the human genome [5,6]. Moreover, genetic screening with the sgRNA library targeting 16,401 lncRNA loci demonstrated cell type specificity and showed that approximately 500 lncRNA loci are required for cellular growth [7]. A next-generation sequencing analysis using the NONCODE database revealed lncRNAs and their relationship with human diseases, exosome expression profiles (such as single nucleotide polymorphism [SNPs]), and secondary structures [8]. Therefore, the role of conserved lncRNAs in both humans and mice has attracted considerable attention in the genomics field. While it is now broadly accepted that lncRNAs cannot merely be considered as “junk,” many questions regarding their role remain in molecular biology [9]. Specifically, the functions of tissue-specific lncRNAs (blood circulation to organs) and their changes in expression are still not clear.

The fundamental function of lncRNAs is the modulation of genetic processes, such as expression and epigenetic modifications at the specific gene locus (Figure 1). Indeed, lncRNAs lack open reading frames (ORFs), and the molecular features of lncRNAs are distinct from those of mRNAs [10]. The role of lncRNAs has been proposed to be an essential transcriptional regulation molecule acting through epigenetic modifications of DNA on specific loci by chromatin remodeling complexes. Subcellular localization of lncRNA is beneficial for studying the biological functions of lncRNAs. The sequence motif-based deep learning method has predicted the distribution of all annotated human lncRNAs. The results have demonstrated the percentage of lncRNAs, such as *MALAT1*, *GAS5*, *NEAT1*, *MEG3*, *HOTAIR*, and *H19* in both fractions, namely, nuclear and cytosol [11]. lncRNAs localized in the cytoplasm have a broad spectrum of functions [12], ranging from the regulation of translation and mitochondrial metabolism [13] to scaffold proteins and sequestration of miRNAs [14,15]. LncRNAs act as cis or trans depending on the transcription sites on the target gene body. Cis-acting lncRNAs are potentially introduced into the genome and chromatin to control the expression of target genes that have been considered to represent a substantial part of known lncRNAs [16]. By contrast, trans-acting lncRNAs regulate gene expression independently of the loci [3,4,17,18]. The cellular hallmarks of lncRNAs interact with various cellular and molecular components like the transcriptional machinery, enhancer RNAs (eRNAs), RNA–chromatin complexes, and targeted chromatin regulators [3,4,17,19,20]. These are achieved by recruiting target molecules and functional localizations, which are also under several levels of control. In addition, it is also required that the exquisite regulation of RNA polymerase allows the buildup of lncRNAs [21].

Traumatic brain injury (TBI) is a physiological brain disorder caused by several unexpected injuries, such as assaults, brain tumors, falls, stroke, and traffic accidents that may all have different outcomes [22]. Clinically, patient outcomes in TBI are tested through cerebrovascular reactivity, and long-term treatment and follow-up are usually required [23]. Positron emission tomography (PET), which visualizes the cerebral arteries in the injured brain, is generally used for tracing or diagnosing TBI progression [24]. Although diagnostic testing for cerebral blood flow (CBF) along with PET was originally used for defining TBI patient outcomes, analysis of damaged brain tissues still shows a high risk for TBI diagnosis. Thus, a mouse model to identify biomarkers in TBI is required for developing efficient diagnostics and therapeutics. Recent SNP studies have enabled the generation of a TBI model that has become a well-appreciated model system for several biological features of TBI [25]. However, mice are less exposed to damage than humans, reflecting some differences in how TBI may manifest in conserved cell types between mice and humans [26]. In addition, stochastic genetic models may represent a paradoxical phenomenon that shows differences between mouse and human TBI.

LncRNAs are a major player in the promotion of targeted gene expression. Considering that they can be bound not only as a secondary or tertiary structure, but also by other RNA-protein complexes [27,28,29], lncRNAs are likely to be involved in many types of human diseases, including TBI. Therefore, in this review, we discussed the main function of lncRNAs in the coordination of several types of TBI, such as stroke and hypoxia, and the possibility of developing new diagnostic and therapeutic targets through the regulation of lncRNAs.

## 2. Stroke

Stroke is a neurological deficit condition caused by a cerebral blood circulation disorder that can lead to death and disabilities. Stroke is estimated to be experienced by one in four adults during their lifetime. Stroke is divided into two main types: Ischemic stroke and hemorrhagic stroke [30]. Ischemic stroke is due to occlusions in vessels (i.e., the carotid artery) resulting in reduction or blockage of blood flow and is responsible for the majority of total stroke incidences worldwide [31]. Hemorrhagic stroke occurs through weakened vessels and vascular rupture and subsequent blood leakage into the brain area, and it can be subcategorized into intracerebral hemorrhage (ICH) and subarachnoid hemorrhage (SAH), depending on the extravasated location [32].

### 2.1. LncRNAs in TBI

The potential correlation between TBI and stroke has been suggested through a cohort analysis of patients with TBI and non-TBI with subsequent stroke. The features of TBI, such as vascular system impairment, affect the blood supply to the brain cells and may trigger ischemic or hemorrhagic stroke [33]. Moreover, 2.5% of patients with moderate to severe TBI were predicted to have an onset of acute ischemic stroke [34]. Recently, the lncRNA profiles in TBI were investigated in the mouse cortex [35] and in whole blood samples derived from TBI patients, leading to the identification of 10 significantly altered expressed lncRNAs [36]. Meanwhile, following TBI injury, various regulatory lncRNAs, such as metastasis-associated lung adenocarcinoma transcript 1 (*MALAT1*), could be used as an approach or delaying secondary injury progression. *MALAT1* is known to have a regenerative role against damage [37]. In this context, treatment with exosome-carried *MALAT1* lncRNA beneficially modulates deficits in TBI models and participates in MAPK pathways [38]. The relative expression level of maternally expressed gene 3 (*MEG3*) lncRNA in plasma from TBI patients was also suggested [39]. The tumor-suppressive *MEG3* lncRNA was found to be significantly downregulated in patients compared to a healthy group, whereas proinflammatory cytokines, including IL-1β, IL-6, IL-8, and TNF-α, were showed upregulated in the plasma of TBI patients. The negative correlation between proinflammatory cytokines and *MEG3* lncRNA was proposed to be used as a tool for evaluating the prognosis of TBI patients [39]. Homeobox (HOX) transcript antisense RNA (*HOTAIR*) lncRNA and lncRNA-p21, which play an oncogenic role [40,41], are also associated with the activation of microglia. Interestingly, lncRNA *Gm4419* increases the expression of TNF-α, leading to apoptosis in astrocytes [42]. In addition, regulation of the inflammatory response molecule, myeloid differentiation factor-88 (MYD88), was reported to be mediated through the inhibition of *HOTAIR* lncRNA and subsequent downregulation of MYD88 [43].

### 2.2. Molecular Features of LncRNAs in Stroke

Recent research has shown that the expression of lncRNAs following stroke displays sexual dimorphism, with 97 differentially expressed lncRNAs found in females and 299 in males in whole blood samples from ischemic stroke patients. Some of these lncRNAs were annotated with ischemic stroke and risk-associated gene loci for other diseases [44]. Further studies identified nine susceptible lncRNAs with respect to ischemic stroke-related deleterious features, such as middle cerebral artery occlusion (MCAO) and/or oxygen-glucose deprivation (OGD)/reperfusion. These lncRNAs were *MALAT1*, *MEG3*, *H19*, *N1LR*, antisense non-coding RNA in the INK4 locus (*ANRIL*), taurine-upregulated gene 1 (*TUG1*), Fos downstream transcript (*FosDT*), CaMK2D-associated transcript 1 (*C2dat1*), and small nucleolar RNA host gene 14 (*SNHG14*), and they were found to have a role in the pathological processes of ischemic stroke, including angiogenesis, inflammatory response, apoptosis, autophagy, and cell death [45]. *ANRIL* has been reported to play a role in thrombosis, which is a critical aspect of occlusion generating-related pathology [46]. A subsequent study using patient-derived plasma suggested that *ANRIL* lncRNA expression has a negative correlation with the severity of ischemic stroke [47]. *H19* lncRNA has been studied in response to hypoxia [48], which is a major cause of ischemic stroke. Upregulation of *H19* lncRNA has been shown to lead to autophagic progress under OGD/reperfusion conditions in vitro via the DUSP5-ERK1/2 axis. In addition, *H19* gene polymorphisms also showed significant differences in specific allele frequency in patients with stroke [49]. 

Recently, *N1LR* lncRNA was shown to attenuate ischemic brain injury by inhibiting the apoptotic process in vivo via the inhibitory effect of *N1LR* against p53 phosphorylation, and was suggested to have an effect on *Nck1*, which is known to be associated with ischemia/reperfusion injury [50]. *C2dat1* is a sense lncRNA against calcium/calmodulin-dependent kinase II δ (CAMK2D/CaMKIIδ), which is an isoform of CaMKII and is activated during glutamate excitotoxicity through massive Ca^2+^ influx. Therefore, *C2dat1* was demonstrated to positively regulate CaMKIIδ following OGD/reperfusion, promoting NF-κB signaling and accounting for focal ischemia in vitro and in vivo [51]. Growth arrest-specific 5 (*GAS5*) is another candidate lncRNA for ischemic stroke pathology, due to its detrimental role in cell survival. *GAS5* was reported to act as a competitive endogenous RNA against miR-137, which protects neurons, in turn lowering cell survival through Notch1 signaling [52]. *MEG3* has also been reported to participate in the mechanism of subarachnoid hemorrhage in hemorrhagic stroke. *MEG3* expression levels in the cerebrospinal fluid were found to be higher in SAH patients than in healthy controls, further promoting neuronal apoptosis via the suppression of the PI3K/AKT pathway [13]. 

Several studies have investigated TBI and stroke separately according to the common lncRNAs between two diseases, including *MALAT1* and *MEG3*. However, the underlying disease mechanisms mediated by lncRNA are still poorly understood, and the evidence is scarce. Therefore, the fundamental role and functions of lncRNAs contributing to both TBI and stroke together need to be elucidated.

## 3. Hypoxia

TBI impairs cerebral perfusion and oxygenation, elevates intracranial pressure, and most of all promotes additional hypoxic/ischemic injuries [53,54]. Hypoxia is a common microenvironmental factor that affects the non-physiological state of oxygen tension. By contrast, normoxia is the level of oxygenation in the healthy state, which is also known as physoxia. Generally, the oxygen concentration in humans varies widely between organs, reflecting metabolic activity and the diversified blood vessel network. In the brain, oxygen concentration ranges approximately 4.6% O_2_, and some neuron types are highly sensitive to hypoxia [55,56,57]. In addition, O_2_ deficiency is already known to lead to the death of cells and tissues. Thus, hypoxia leads to various disorders in the human brain that can cause nerve cell damage [58]. Hypoxic brain injury is the most common cause of neurobehavioral and cognitive dysfunction and subsequent death. Also, hypoxic brain injury promotes various central nervous system diseases, and the brain hypoxia-ischemia lead to brainstem, and cerebellum has been revealed to enhance neurological injury [59]. Hypoxic-ischemic brain damage (HIBD) causes neuronal death and leads to various human neurological dysfunctions, including learning disabilities, movement disorders, cerebral palsy, epilepsy, and even death by participating in various apoptosis-related signaling pathways [60].

### 3.1. Influence of LncRNAs in Neuronal Cell Fate

Hypoxia-inducible factors (HIFs) are one of the most common regulators of O_2_ homeostasis. HIFs modulate various biological processes as a transcriptional activator through the transactivation domains and oxygen-dependent degradation domain. HIF-1 plays an important role in response to hypoxia and ischemia [58]. Also, members of the hypoxia-inducible factor family, HIF-1α that have other alpha subunits, including HIF-2α and HIF-3α as the structural and biochemical similarities [61]. Furthermore, lncRNAs were revealed to interact with HIFs that play important roles in various human diseases [62,63,64,65]. Recent studies have demonstrated that HIF-1 is closely related to HIBD and TBI as the key regulation of factors. [66,67,68,69]. Also, several studies have reported that lncRNAs have a crucial role in hypoxia/ischemic diseases involving HIBD via regulating the expression of target genes [64,70,71,72,73]. In neonatal rat brains, hypoxic-ischemic brain damage was found to change the expression of lncRNAs, such as *ENSRNOG00000021987*, which is downregulated in HIBD. Also, *ENSRNOG00000021987* promoted neuronal apoptosis-induced HIBD via regulating the apoptosis-related genes and proteins [74]. The lncRNA *BC088414* was found to have an altered high expression under the hypoxic/ischemic brain injury in the neonatal rat brain. Knockdown of lncRNA *BC088414* decreased the mRNA expression of caspase-6 (Casp6) and β2-adrenoceptor (Adrb2), which are apoptosis-related genes, thereby reducing the cell apoptosis and promoting cell proliferation in pheochromocytoma-12 (PC-12) neural cells. Thus, lncRNA may contribute to the pathogenesis of HIBD through the regulation of coding genes [75,76]. The lncRNA growth arrest-specific 5 (*GAS5*) is overexpressed after HIBD. Silencing *GAS5* participates in the protection against HIBD via regulating the hippocampal neuron function through the sponged miR-23a [77]. In addition, *GAS5* directly binds to miR-221 as a competitive endogenous RNA (ceRNA) and regulates the p53-upregulated modulator of apoptosis (PUMA), which is required for apoptosis as a member of the Bcl-2 protein family and the target of miR-211, thereby enhancing the neuronal apoptosis under hypoxia [14]. The lncRNA small nucleolar RNA host gene 1 (*SNHG1*) is downregulated under hypoxia conditions in neuroblastoma, SH-SY5Y cells, inhibiting cell viability and promoting cell apoptosis through the suppression of Bcl-XL as an anti-apoptotic protein via targeting miR-140-5p. Therefore, *SNHG1* mediated miR-140-5p-Bcl-XL axis, may provide protection from hypoxic brain injury [78].

lncRNA *H19* is upregulated under hypoxia conditions. High expression of *H19* induces neuroinflammation and autophagy in ischemic stroke [49,79]. In addition, silenced *H19* protects the neural cell, PC-12, under hypoxia-induced injury via sponging the miR-28, thereby inhibiting SP1 form degradation that promotes the deactivation of PDK/AKT and JAK/STAT signaling pathways [80]. Besides, *H19* is a dual competitive interaction of miR-19a and inhibitor of DNA binding/differentiation 2 (Id2) that promotes hypoxic-ischemic neuronal apoptosis. Therefore, inhibition of the *H19*-miR-19a-Id2 axis protects against hypoxic-ischemic neuronal injury. Therefore, the lncRNA *H19* implies the key regulator participating in the pathogenesis of hypoxic brain damage [15]. The lncRNA nuclear-enriched abundant transcript 1 (*NEAT1*) has been reported to contribute to traumatic brain injury recovery via suppressing the inflammation and cell apoptosis through the overexpression of *NEAT1* [81]. *NEAT1* promotes the expression of homeobax A1 (HOXA1), which modulates diverse biological processes involving neurogenesis through the gene expression, to reduce the neuronal cell apoptosis via sponging the miR-339-5p as the ceRNA. Thereby, it attenuates HIBD progression [82]. The lncRNA metastasis-associated lung adenocarcinoma transcript 1 (*MALAT1*) is upregulated under hypoxia conditions in PC-12 cells, which leads to activation of the p38 MAPK signaling pathway as the promotion of cell apoptosis and reduction of cell viability. Therefore, suppression of *MALAT1* may protect neurons from HIBD [83]. The expression of lncRNA urothelial carcinoma-associated 1 (*UCA1*) is promoted under hypoxia. Downregulated *UCA1* upregulates the miR-18a, which inhibits the SRY-box containing gene 6 (SOX6) as the target gene, thus, preventing hypoxia injury following cerebral ischemia via increasing migration, invasion, and viability, and inhibiting apoptosis in neural PC-12 cells [84]. The lncRNA *TCONS_00044595* has elevated the expression in the pineal gland and striatum, which reveals a circadian pattern within the pineal gland. TCONS_00044595 as a ceRNA of miR-182 and regulates the pineal CLOCK as the circadian regulator, thereby associating with the abnormal changes in the pineal gene upon HIBD [85]. The lncRNA myocardial infarction-associated transcript (*MIAT*) is downregulated under hypoxic/ischemic injury in the striatal tissues of neonatal rats and Neuro2A cells. However, overexpression of *MIAT* suppresses the apoptosis via interacting with miR-211 and glial cell line-derived neurotrophic factor (GDNF), which repairs and protects the neurons, thereby attenuating the hypoxic/ischemic injury in neonatal rats [86]. The lncRNA *ROR* binds and suppresses the miR-135a-5p as a ceRNA, which upregulates the expression of ROCK1/2. ROCK1/2 is associated with caspase-related apoptosis as the pyrolysis product of activated caspase 3 and caspase 2 [87,88]. Therefore, the *ROR* promotes the cerebral hypoxia/reoxygenation-induced injury in PC-12 cells [89].

### 3.2. Possibilities of LncRNA for Targeted Therapy for TBI

Many hypoxia and ischemia studies have reported the observation that vascular response deficiency often leads to increased mutations, both of which have been correlated with a vascular response in a variety of transcriptional regulation process [90,91]. However, the relationship between post-transcriptional mechanisms under lncRNA expression and vascular response in TBI is more complex. It is likely that studies of lncRNAs in hypoxic conditions will continue to reveal associations between transcriptional regulation, due to mutation or expression of genes in TBI. Although lncRNAs promote durable responses in several hypoxic conditions, the response rates and in vivo significance among TBI patients need to be clarified. Therefore, major challenges remain in identifying predictive biomarkers for hypoxia-induced TBI that can guide the use of several anti-hypoxia agents as a combination treatment strategy involving lncRNA-targeted therapies. In addition, the principles that have been used to identify new biomarkers for the response to hypoxic/ischemic injury will likely be applicable for diagnosis or emerging therapies.

## 4. Further Research Directions

An accurate genomic mechanism mediated by post-transcriptional regulation is a hallmark of neuronal differentiation in mature neurons. Recently, it has become clear that lncRNAs have a major impact on transcriptional regulation, with important therapeutic implications in various brain diseases. Although lncRNAs are involved in general responses in many patients with TBI, significant expression changes or mutations may show individually. Thus, a major challenge remains to identify biomarkers that can improve combined treatment strategies involving targeted TBI therapies. As we have reviewed here, many studies of lncRNAs in brain injuries conclude that mutation, depletion, and overexpression lead to pathogenicity, which has been correlated with TBI response in a variety of settings.

Although lncRNA targeted therapies are rapidly reshaping the treatment of brain diseases [92], some RNA-targeted oligonucleotides using complementary sequences have already been approved in the clinic, and numerous additional approvals are foreseen in the future [93]. However, the relationship between lncRNAs and the response to TBI is more complex. The source and identity of lncRNAs that affect neural cell development leading to the clinical benefits of lncRNAs, have not been fully characterized. Therefore, such lncRNAs may be critical for TBI treatments that involve the transcriptional machinery in brain injury. Various lncRNAs are associated with brain injury response in biochemical experimental results (Figure 2). The most powerful screening tools for the identification of lncRNAs will likely reveal the multiple types of TBI. In addition, there is much additional evidence regarding the regulatory functions of lncRNAs in TBI. [94].

Although the development of successful small-molecule therapeutics for TBIs has been delayed owing to various technical obstacles, highly structured lncRNAs, and their regulatory proteins have been demonstrated as the targets of small molecules or antisense oligonucleotides (ASO) for TBI therapy [95]. Considering the lncRNA activity and its structural modules, several studies have provided evidence for lncRNA-targeted therapy. For instance, the impairment of lncRNA MED3 by pseudoknot structures modulates p53 stimulation [28]. Evolutionarily conserved lncRNAs, such as *HOTAIR*, *MALAT1*, *GAS5*, and *NEAT1*, which form multiple structures, also play a crucial role in promoting a variety of cellular processes [96,97,98,99]. These observations suggest that lncRNA can be a novel target for the development of small-molecule drugs. As ASOs are associated with the regulation of protein expression via RNA modulation, they have been demonstrated as novel chemotherapy drugs [100]. Recently, a study showed that lncRNA *TUG1* may be a potential target for glioma treatment with ASO [101]. The results demonstrated the utility of ASOs in the inhibition of lncRNAs in TBIs with certain drugs [101]. Additionally, eRNAs comprise one of the lncRNA families that are transcribed from active enhancer elements and specifically regulate gene transcription and translation via NELF release from the target gene promoter in cortical neurons [3]. A recent study has suggested that eRNA has a high potential for clinical use in cancer therapy. NET1-associated eRNA (NET1e), which promotes cell proliferation in cancer cells, significantly decreased cell proliferation by suppressing NET1e transcription using the locked nucleic acid (LNA) method [102]. Therefore, lncRNA-targeted therapy with small-molecule drugs and ASOs may be a novel and powerful clinical approach for the treatment of TBIs.

We have reviewed important insights into the roles of lncRNAs as critical players in TBI as revealed through genetic and biochemical studies (Table 1). However, many unanswered questions remain. How is mutated DNA transcribed to lncRNAs without splicing steps to then show individual-specific phenotypes in TBI? What are the recruitment timing and localization of lncRNAs on chromosomes in the nucleus and the cytosol during the progression of TBI? How many lncRNA-specific loci and mutations exist in TBI tissues? These and other emerging questions may uncover even more interesting and critical features of lncRNAs.

## 5. Conclusions

Conventional understanding states that gene mutations aggravate TBI through the association of injury repair, vascular regeneration, and neuroinflammation, and autophagy. However, it is difficult to determine the relative contribution of familial variants in TBIs and those that originate owing to the exposure to brain damage in different patients. Most mutations may not be related to TBI progression. Therefore, studies that reveal the correlation between genetic variation and their associated mechanisms of action, such as the types of novel lncRNAs (eRNA, mRNA-flanking lncRNAs, etc.), may have the potential to better define the understanding of TBI risk during treatment, as well as type-specific and individual risks. In addition, strong preclinical background on both the genotype and phenotype of lncRNAs in TBIs is essential. Predictive mutations of genetic risk, which are rigorously preclinically validated through various assays, may be utilized in the diagnosis of TBIs.

Current technical developments, such as single-cell based next-generation sequencing (NGS) and genome-wide association study (GWAS), for understanding the basic regulation of lncRNAs have led to advances in transcriptome-based drug development [103]. These methods have identified tens of thousands of lncRNA loci, which range from single cells to tissues and are disease-specific in humans. For several decades, significant efforts have been focused on identifying unique lncRNAs that could serve as biomarkers of brain injuries. Large-scale genomic studies have begun to reveal the mutational status individually in brain disorders, and it is now possible to rationally select SNP-based mutations in brain disorders. This genetic concept, combined with the roles of lncRNAs as a non-coding entity able to control the whole chromosome, provides a framework for the design of novel therapeutic approaches. In summary, lncRNAs function as molecular orchestral conductors during transcriptional events by sequentially calling different states into action in TBI; these lncRNAs are associated with the progression of TBI, due to their diverse features. Further investigation of brain injury-specific lncRNAs is of crucial importance for scientists to unveil more details about the influence of the transcriptional machinery in various brain disorders.

## Figures and Tables

**Figure 1 biology-09-00458-f001:**
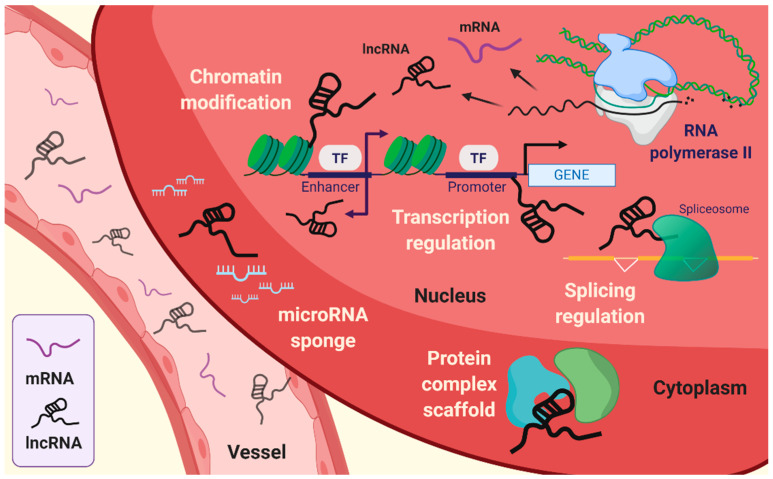
Emerging roles of long non-coding RNAs (lncRNAs). lncRNAs are associated with intracellular mechanisms, such as transcriptional regulation, splicing regulation, choromatin modification, and scaffolding of protein complex. In addition, lncRNAs can sequestrate microRNA by sponging.

**Figure 2 biology-09-00458-f002:**
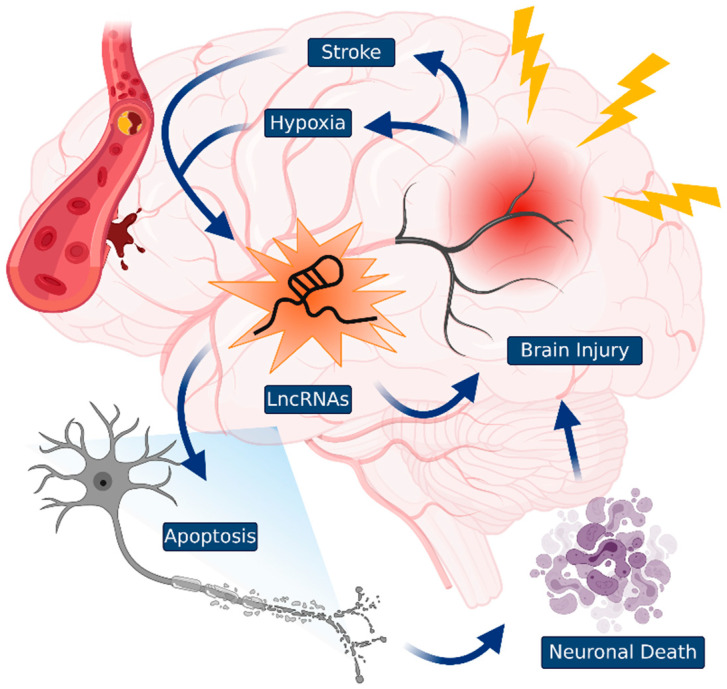
Traumatic brain injuries (TBIs) and lncRNAs. TBIs that occurred by diverse damage to the brain induces a cascade of neuronal cell death events. The specific lncRNAs control TBIs, such as stroke and hypoxia. Identification of novel lncRNAs could be useful as a guide to TBI diagnosis and may be used in the development of novel therapeutic targets.

**Table 1 biology-09-00458-t001:** A list of lncRNAs in brain injury.

Brain Injury	lncRNA	Targets	Function	Ref.
TBI	*MALAT1*	MAPK pathways	Beneficially regulates deficits in TBI models	[38]
	*MEG3*	IL-1β, IL-6, IL-8, and TNF-α	Upregulates pro-inflammatory cytokines in the plasma of TBI patients	[39]
	*p21*		Microglial activation	[41]
	*Gm4419*	TNF-α	Astrocyte apoptosis	[42]
	*HOTAIR*	MYD88	Activates the microglia and regulates the inflammatory response molecule	[43]
Stroke	*MALAT1*		Contributes to the pathological process of ischemic stroke	[45]
	*TUG1*			
	*FosDT*			
	*SNHG14*			
	*ANRIL*		Roles the negative correlation with the severity of ischemic stroke	[45,47]
	*H19*		Contributes to the pathological process of ischemic stroke and shows differences in specific allele	[45,49]
	*N1LR*	p53 and Nck1	Attenuates ischemic brain injury by inhibiting the apoptotic process and associates the pathological process of ischemic stroke	[45,50]
	*C2dat1*	CaMKIIδ	Modulates CaMKIIδ under oxygen-glucose deprivation (OGD)/reperfusion via promoting NF-κB signaling and accounting for focal ischemia in vitro and in vivo	[45,51]
	*GAS5*	miR-137	Inhibits the protection of neurons and decreases cell viability through the Notch1 signaling	[52]
	*MEG3*		Participates in pathological process of hemorrhagic and ischemic stroke and enhances neuronal apoptosis via the suppression of the PI3K/AKT pathway	[13,45]
Hypoxia	*ENSRNOG* *00000021987*		Promote neuronal apoptosis induced hypoxic-ischemic brain damage (HIBD) via increasing apoptosis-related genes and proteins	[74]
	*BC088414*	Casp6 and Adrb2	Reduces PC-12 cell apoptosis and increases proliferation via decreasing apoptosis-related gene	[75,76]
	*GAS5*	miR-23a	Protects against HIBD via modulating hippocampal neuron function	[77]
		miR-221/PUMA	Promotes the neuronal apoptosis under hypoxia	[14]
	*SNHG1*	miR-140-5p /Bcl-XL	Inhibits neuroblastoma cell viability and enhances cell apoptosis under hypoxia	[78]
	*H19*	miR-28/SP1	Protects from the hypoxia-induced injury in PC-12 cell via deactivating the PDK/AKT and JAK/STAT pathways	[80]
		miR-19/Id2	Promotes the hypoxic-ischemic neuronal apoptosis	[15]
	*NEAT1*	miR-339-5p /HOXA1	Attenuates HIBD progression via reducing neuronal cell apoptosis	[82]
	*MALAT1*	p38 MAPK pathway	Enhances the cell apoptosis and decreases cell viability under hypoxia in PC-12	[83]
	*UCA1*	miR-18a/SOX6	Prevents hypoxia injury following cerebral ischemia via promoting migration, invasion, and viability, and reducing apoptosis in PC-12	[84]
	*TCONSA_* *00044595*	miR-182/CLOCK	Relates to abnormal changes in the pineal gene upon HIBD	[85]
	*MIAT*	miR-211/GDNF	Suppresses hypoxic/ischemic injury in neonatal rat via inhibiting neurons apoptosis	[86]
	*ROR*	miR-135a-5p/ ROCK1/2	Promotes cerebral hypoxia/reoxygenation-induced injury via increasing caspase-related apoptosis in PC-12	[89]

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
