# Peer review of "Discoveries for Long Non-Coding RNA Dynamics in Traumatic Brain Injury"

_biology, 2020, doi:10.3390/biology9120458_

Round 1
Reviewer 1 Report
In general, the material is well organized and easy to be followed and understood.
The tables and the graphics are informative and very well organized.
A few minor remarks are listed below:
Line 30: lncRNAs are longer than 200nt.
Line 42: I don't think that "repression" is the proper word here.
Line 126: "Via" should not be in Italic
Author Response
Reviewer #1
In general, the material is well organized and easy to be followed and understood.
The tables and the graphics are informative and very well organized.
We thank the reviewer and corrected this manuscript by the reviewer’s comments.
A few minor remarks are listed below:
Comment #1. Line 30: lncRNAs are longer than 200nt.
Answer: We have corrected the description in the revised manuscript (Please refer to Line 29).
Comment #2. Line 42: I don't think that "repression" is the proper word here.
Answer: We have corrected the word in the revised manuscript (Please refer to Line 42).
Comment #3. Line 126: "Via" should not be in Italic.
Answer: We have corrected the description in the revised manuscript (Please refer to Line 134, 138, and 149).

Reviewer 2 Report
The manuscript by Lim KH and co-authors provide a comprehensive review on the role lncRNAs in traumatic brain injury (TBI). The article provides a perspective on pathophysiology of lncRNAs in brain stroke, hypoxia and injury repair. The non-exhaustive list of the lncRNAs implicated in TBI can serve as potential diagnostic markers and therapeutic targets. The manuscript is well-written, but may benefit from considering the points listed below:
- The manuscript needs to be edited for grammar. Certain sentences in the abstract and future research directions sections require rewriting to coherently convey the information.
- The description of lncRNAs in line 29 is incorrect. lncRNAs are greater that 200 nucleotides (NOT shorter).
- The reference to figure 2 in line 42 is incorrect. It should be figure 1.
- The authors should include references for cytoplasmic functions of lncRNAs. Although authors have indicated this in figure 1, they have not listed nor provided citations for lncRNAs role in cytoplasm viz. mRNA turnover, proteome homeostasis, sequestering of cytosolic factors etc. This is relevant to the review as lncRNA GAS5, H19, MALAT1 and some others implicated in TBI have functions in the cytoplasm.
- While the authors have provided a comprehensive list of lncRNAs and target signaling pathways, they do not discuss the role of these pathways in brain injury. A brief discussion of the contributions of these pathways in injury repair, vascular regeneration, neuroinflammation and autophagy will provide a better perspective and understanding of lncRNA functions to the readers.
- The authors in the introduction section compare the human and mouse models used to study TBI. The manuscript will benefit from the discussion of the TBI models, both in vivo and in vitro, used to investigate lncRNA functions in TBI. Further, the manuscript will also benefit from including the information of the model (human, mouse or in vitro) used to explore the role on certain specific lncRNAs. For example, the mouse BV2 cells (not mentioned in the manuscript) were used to understand the mechanistic contribution of lncRNA HOTAIR in microglia activation.
- The authors have not included some recently identified lncRNAs implicated in TBI. The authors have not discussed the role of lncRNAs p21 (in microglial activation), Gm4419 (in apoptosis) and CRNDE (in neuroinflammation) among others in TBI.
- The authors should expand on the strategies of ‘lncRNA targeted therapies’ mention in the future research directions section (line 253). A discussion of recent work on ASO based therapies, using modified oligonucleotides, to target TUG1 and other oncogenic lncRNAs, also implicated in TBI, would add more information to this section.
Author Response
Reviewer #2
The manuscript by Lim KH and co-authors provide a comprehensive review on the role lncRNAs in traumatic brain injury (TBI). The article provides a perspective on pathophysiology of lncRNAs in brain stroke, hypoxia and injury repair. The non-exhaustive list of the lncRNAs implicated in TBI can serve as potential diagnostic markers and therapeutic targets. The manuscript is well-written, but may benefit from considering the points listed below:
We thank the reviewer for the careful reading and pointing out the lack of information in the manuscript.
Comment #1. The manuscript needs to be edited for grammar. Certain sentences in the abstract and future research directions sections require rewriting to coherently convey the information.
Answer: We have a thorough language editing for the manuscript to ensure that such grammatical and logical mistakes are avoided.
Comment #2. The description of lncRNAs in line 29 is incorrect. lncRNAs are greater that 200 nucleotides (NOT shorter).
Answer: We have corrected the description in the revised manuscript (Please refer to Line 29).
Comment #3. The reference to figure 2 in line 42 is incorrect. It should be figure 1.
Answer: We have corrected the word in the revised manuscript (Please refer to Line 29).
Comment #4. The authors should include references for cytoplasmic functions of lncRNAs. Although authors have indicated this in figure 1, they have not listed nor provided citations for lncRNAs role in cytoplasm viz. mRNA turnover, proteome homeostasis, sequestering of cytosolic factors etc. This is relevant to the review as lncRNA GAS5, H19, MALAT1 and some others implicated in TBI have functions in the cytoplasm.
Answer: We have added the description in the revised manuscript (Please refer to Line 46-52).
Comment #5. While the authors have provided a comprehensive list of lncRNAs and target signaling pathways, they do not discuss the role of these pathways in brain injury. A brief discussion of the contributions of these pathways in injury repair, vascular regeneration, neuroinflammation and autophagy will provide a better perspective and understanding of lncRNA functions to the readers.
Answer: We highly appreciate this suggestion and have now added a description of the in Conclusion of our revised manuscript (Please refer to Line 304-313).
Comment #6. The authors in the introduction section compare the human and mouse models used to study TBI. The manuscript will benefit from the discussion of the TBI models, both in vivo and in vitro, used to investigate lncRNA functions in TBI. Further, the manuscript will also benefit from including the information of the model (human, mouse or in vitro) used to explore the role on certain specific lncRNAs. For example, the mouse BV2 cells (not mentioned in the manuscript) were used to understand the mechanistic contribution of lncRNA HOTAIR in microglia activation.
Answer: We thank the reviewer for the positive assessment of our manuscript.
Comment #7. The authors have not included some recently identified lncRNAs implicated in TBI. The authors have not discussed the role of lncRNAs p21 (in microglial activation 112), Gm4419 (in apoptosis 115) and CRNDE (in neuroinflammation) among others in TBI.
Answer: We thank the reviewer for the critical information. We have added the description and modified the Table.1 in the revised manuscript (Please refer to Line 112-115, and Table.1) We have carefully considered with reviewer’s concern for adding CRNDE. Because CRNDE was only showed in pain behaviors with neuroinflammation, it is likely out of scope for TBIs. We therefore did not add CRNDE in a list of TBI related lncRNA.
Comment #8. The authors should expand on the strategies of ‘lncRNA targeted therapies’ mention in the future research directions section (line 253). A discussion of recent work on ASO based therapies, using modified oligonucleotides, to target TUG1 and other oncogenic lncRNAs, also implicated in TBI, would add more information to this section.
Answer: We thank the reviewer for excellent point and this concern also raised by the reviewer #3. We have added more information in the revised manuscript (Please refer to Line 272-291).

Reviewer 3 Report
Lim et al revise the literature covering the roles of lncRNAs in brain injuries. Their report is sound and useful for a specific targeted audience working on lncRNA-based medicine, which is a rapidly expanding field.
I recommend publication of their review, pending the following corrections:
page 1 line 30: an annotation of lncRNA databases is summarized in a recent review, which should be cited here: Chillon I., Marcia M. (2020) The molecular structure of long non-coding RNAs: emerging patterns and functional implications. Crit Rev Biochem Mol Biol 55:662-690.
page 1 line 38: "junk". An update on the concept of lncRNAs as junk transcripts has recently been published and should be cited here: Palazzo A. F., Koonin E. V. (2020) Functional Long Non-coding RNAs Evolve from Junk Transcripts. Cell 183:1151-1161
page 2 line 74: the sentence "Considering that they can be bound not only as a secondary or tertiary structure but also by other RNA-protein complexes" is unclear. Moreover, for the role of lncRNA tertiary structure the following references should be cited:
Uroda T., Anastasakou E., Rossi A., Teulon J. M., Pellequer J. L., Annibale P., Pessey O., Inga A., Chillon I., Marcia M. (2019) Conserved Pseudoknots in lncRNA MEG3 Are Essential for Stimulation of the p53 Pathway. Mol Cell 75:982-995 e989
Kim D. N., Thiel B. C., Mrozowich T., Hennelly S. P., Hofacker I. L., Patel T. R., Sanbonmatsu K. Y. (2020) Zinc-finger protein CNBP alters the 3-D structure of lncRNA Braveheart in solution. Nat Commun 11:148.
page 3 line 100: the sentence "The relative expression level of maternally expressed gene 3 (MEG3) lncRNA in plasma from TBI patients was also suggested." is unclear
chapters 2.2 and 3.1, and Table 1. Several of the lncRNAs mentioned here are highly structured lncRNAs. Therefore, I would recommend adding a paragraph in section 3.2 or in section 4 to highlight this important property of lncRNAs connected to brain injuries and to propose that these lncRNAs - thanks to the fact that they are highly structured - they could perhaps in the future be targeted by small molecules, as a new therapy against brain injury. See (and cite as appropriate):
MEG3: Uroda T., Anastasakou E., Rossi A., Teulon J. M., Pellequer J. L., Annibale P., Pessey O., Inga A., Chillon I., Marcia M. (2019) Conserved Pseudoknots in lncRNA MEG3 Are Essential for Stimulation of the p53 Pathway. Mol Cell 75:982-995 e989.
HOTAIR: Somarowthu S., Legiewicz M., Chillon I., Marcia M., Liu F., Pyle A. M. (2015) HOTAIR forms an intricate and modular secondary structure. Mol Cell 58:353-361.
MALAT1: McCown P. J., Wang M. C., Jaeger L., Brown J. A. (2019) Secondary Structural Model of Human MALAT1 Reveals Multiple Structure-Function Relationships. Int J Mol Sci 20.
GAS5: Frank F., Kavousi N., Bountali A., Dammer E. B., Mourtada-Maarabouni M., Ortlund E. A. (2020) The lncRNA Growth Arrest Specific 5 Regulates Cell Survival via Distinct Structural Modules with Independent Functions. Cell Rep 32:107933.
NEAT1: Lin Y., Schmidt B. F., Bruchez M. P., McManus C. J. (2018) Structural analyses of NEAT1 lncRNAs suggest long-range RNA interactions that may contribute to paraspeckle architecture. Nucleic Acids Res 46:3742-3752.
page 6 line 254: the sentence "some RNA-targeted oligonucleotides using complementary sequences have already been approved in the clinic and numerous additional approvals are foreseen in the future" needs to be supported by a reference.
Author Response
Reviewer #3
Lim et al revise the literature covering the roles of lncRNAs in brain injuries. Their report is sound and useful for a specific targeted audience working on lncRNA-based medicine, which is a rapidly expanding field.
We would like to thank the reviewer for careful reading and suggesting with recent studies.
I recommend publication of their review, pending the following corrections:
Comment #1. page 1 line 30: an annotation of lncRNA databases is summarized in a recent review, which should be cited here: Chillon I., Marcia M. (2020) The molecular structure of long non-coding RNAs: emerging patterns and functional implications. Crit Rev Biochem Mol Biol 55:662-690.
Answer: We have cited the reference in the revised manuscript (Please refer to Line 31).
Comment #2. page 1 line 38: "junk". An update on the concept of lncRNAs as junk transcripts has recently been published and should be cited here: Palazzo A. F., Koonin E. V. (2020) Functional Long Non-coding RNAs Evolve from Junk Transcripts. Cell 183:1151-1161
Answer: We have cited the reference in the revised manuscript (Please refer to Line 38).
Comment #3. page 2 line 74: the sentence "Considering that they can be bound not only as a secondary or tertiary structure but also by other RNA-protein complexes" is unclear. Moreover, for the role of lncRNA tertiary structure the following references should be cited: o o
Uroda T., Anastasakou E., Rossi A., Teulon J. M., Pellequer J. L., Annibale P., Pessey O., Inga A., Chillon I., Marcia M. (2019) Conserved Pseudoknots in lncRNA MEG3 Are Essential for Stimulation of the p53 Pathway.
Kim D. N., Thiel B. C., Mrozowich T., Hennelly S. P., Hofacker I. L., Patel T. R., Sanbonmatsu K. Y. (2020) Zinc-finger protein CNBP alters the 3-D structure of lncRNA Braveheart in solution. Nat Commun 11:148.
Answer: We have modified and cited the reference in the revised manuscript (Please refer to Line 80-81).
Comment #4. page 3 line 100: the sentence "The relative expression level of maternally expressed gene 3 (MEG3) lncRNA in plasma from TBI patients was also suggested." is unclear
Answer: We have added the reference to clarify our suggestion in the revised manuscript (Please refer to Line 108).
Comment #5. chapters 2.2 and 3.1, and Table 1. Several of the lncRNAs mentioned here are highly structured lncRNAs. Therefore, I would recommend adding a paragraph in section 3.2 or in section 4 to highlight this important property of lncRNAs connected to brain injuries and to propose that these lncRNAs - thanks to the fact that they are highly structured - they could perhaps in the future be targeted by small molecules, as a new therapy against brain injury.
See (and cite as appropriate):
MEG3: Uroda T., Anastasakou E., Rossi A., Teulon J. M., Pellequer J. L., Annibale P., Pessey O., Inga A., Chillon I., Marcia M. (2019) Conserved Pseudoknots in lncRNA MEG3 Are Essential for Stimulation of the p53 Pathway. Mol Cell 75:982-995 e989.
HOTAIR: Somarowthu S., Legiewicz M., Chillon I., Marcia M., Liu F., Pyle A. M. (2015) HOTAIR forms an intricate and modular secondary structure. Mol Cell 58:353-361.
MALAT1: McCown P. J., Wang M. C., Jaeger L., Brown J. A. (2019). Int J Mol Sci 20.
GAS5: Frank F., Kavousi N., Bountali A., Dammer E. B., Mourtada-Maarabouni M., Ortlund E. A. (2020) The lncRNA Growth Arrest Specific 5 Regulates Cell Survival via Distinct Structural Modules with Independent Functions. Cell Rep 32:107933.
NEAT1: Lin Y., Schmidt B. F., Bruchez M. P., McManus C. J. (2018) Structural analyses of NEAT1 lncRNAs suggest long-range RNA interactions that may contribute to paraspeckle architecture. Nucleic Acids Res 46:3742-3752.
Answer: We thank to the reviewer for suggestion and added a paragraph in the revised manuscript (Please refer to Line 272-291).
Comment #6. page 6 line 254: the sentence "some RNA-targeted oligonucleotides using complementary sequences have already been approved in the clinic and numerous additional approvals are foreseen in the future" needs to be supported by a reference.
Answer: We have added the reference in the revised manuscript (Please refer to Line 264).
